# Control of PNG kinase, a key regulator of mRNA translation, is coupled to meiosis completion at egg activation

**Masatoshi Hara[1][†], Boryana Petrova[1], Terry L Orr-Weaver[1,2]\***

[1]Whitehead Institute and Massachusetts Institute of Technology, Cambridge, United States; [2]Department of Biology, Massachusetts Institute of Technology, Cambridge, United States

**Abstract** The oocyte-to-embryo transition involves extensive changes in mRNA translation, regulated in Drosophila by the PNG kinase complex whose activity we show here to be under precise developmental control. Despite presence of the catalytic PNG subunit and the PLU and GNU activating subunits in the mature oocyte, GNU is phosphorylated at Cyclin B/CDK1sites and unable to bind PNG and PLU. In vitro phosphorylation of GNU by CyclinB/CDK1 blocks activation of PNG. Meiotic completion promotes GNU dephosphorylation and PNG kinase activation to regulate translation. The critical regulatory effect of phosphorylation is shown by replacement in the oocyte with a phosphorylation-resistant form of GNU, which promotes PNG-GNU complex formation, elevation of Cyclin B, and meiotic defects consistent with premature PNG activation. After PNG activation GNU is destabilized, thus inactivating PNG. This short-lived burst in kinase activity links development with maternal mRNA translation and ensures irreversibility of the oocyte-to-embryo transition.

**\*For correspondence:** weaver@ wi.mit.edu

**Present address:** [†]Graduate School of Frontier Biosciences, Osaka University, Suita, Japan

**Competing interests:** The authors declare that no competing interests exist.

## Introduction

The onset of embryogenesis at the oocyte-to-embryo transition occurs in the absence of transcription and thus requires remodeling of mRNA translation that alters the proteome (*Tadros and Lipshitz, 2009*). This dramatic alteration in maternal mRNA translation is timed to be coordinated with egg activation and the completion of meiosis, but the mechanisms linking these developmental events are unknown. The production of a mature oocyte and its transition to an embryo involve arrest and release of meiosis at two points (*Von Stetina and Orr-Weaver, 2011*). The immature oocyte is arrested at prophase of meiosis I (prophase I) in nearly all animals to permit oocyte differentiation, and this arrest is released as oocyte maturation initiates. Maturation leads to the oocyte entering the meiotic divisions and arresting at a second point, metaphase of meiosis II (metaphase II) in vertebrates and metaphase of meiosis I (metaphase I) in insects.

We use the term egg activation to denote the onset of events that mark the transition of oocyte to embryo, such as release of the secondary meiotic arrest, completion of meiosis, and changes in translation of maternal mRNAs. In many organisms egg activation is linked to fertilization, although it can be independent in organisms such as Drosophila (*Horner and Wolfner, 2008b*). The profound developmental change as the oocyte transitions to an embryo involves extensive reorganization of the proteome. In contrast to somatic cells, in which transcription is a major source of gene expression changes, in most animal species oocytes and early embryos are dependent on posttranscriptional control (*Walker et al., 1996*; *Potireddy et al., 2006*; *Chen et al., 2011*; *Krauchunas and Wolfner, 2013*; *Robertson and Lin, 2013*; *Kronja et al., 2014a*, *2014b*). The absence of

**eLife digest** New egg cells form via a specialized kind of cell division called called meiosis, and will pause at key stages in this process before continuing their development. One of these pauses occurs before the egg cell is fertilized. At fertilization, the egg cell becomes "activated", development resumes, and it starts forming into an embryo. Molecules deposited in the egg cell when it originally formed are used to control these earliest stages of embryonic development. These molecules include messenger RNA molecules (mRNAs for short), which can be "translated" to build proteins.

In fruit flies, an enzyme called PNG kinase regulates the translation of hundreds of mRNA molecules during the period after the pause, when the maturing egg cell is activated and the embryo begins to develop. It is not well understood what activates and inactivates the kinase to limit its activity to this period of time. However, it was known that a protein called GNU was needed to bind to the PNG kinase to make it active.

CyclinB/CDK1 is another kinase, and in contrast to PNG it is highly active when the egg cell is paused. When the egg cell is activated for embryonic development, the levels of this second kinase drop sharply and meiosis is completed. Like all kinases, CyclinB/CDK1 attaches phosphate groups onto other molecules, and Hara et al. now show that CyclinB/CDK1 can modify the GNU protein in this way. The added phosphate groups prevent GNU from binding to the PNG kinase, meaning that the high levels of CyclinB/CDK1 during the pause stop GNU from activating the PNG kinase. However, when the egg cell is activated, the level of CyclinB/CDK1 declines so that there are not enough of these molecules to add phosphates onto GNU. This leaves GNU free to activate the PNG kinase, allowing this kinase to control the translation of mRNA molecules. Furthermore, the activity of PNG kinase leads to the destruction of GNU, and this feedback loop limits this kinase's activity to the narrow window of time in which it is needed.

The fruit fly is the second example of an animal in which the activity of a kinase essential for embryonic development has been linked to the completion of meiosis (the other being the roundworm *Caenorhabditis elegans).* The use of this strategy in two such different animals suggests that it may also be common to many other animals, including humans. Further investigation is now needed to determine if this is indeed the case.

transcription in this developmental phase necessitates translational control of maternal mRNAs deposited in the oocyte. Changes in protein stability also influence the proteome during this transition.

A key question in developmental biology is how these changes in translation and protein stability are controlled. The role of translation in the onset of oocyte maturation has been studied in vertebrates (*Mendez and Richter, 2001*; *Chen et al., 2011*, *2013*), but translational control during the oocyte-to-embryo transition is poorly understood. In addition, the events accompanying egg activation must be coupled to the completion of meiosis, but the mechanistic basis of this linkage is unknown. To define how changes in translation are coordinated with meiotic cell cycle progression following egg activation, we use the fruit fly, *Drosophila melanogaster.*

In Drosophila, an oocyte develops through fourteen morphologically distinct stages in an egg chamber consisting of fifteen nurse cells and surrounding follicle cells (*Spradling, 1993*). Oocytes in early stages (from stage 5 to 12) are arrested at prophase I. Oocytes in stage 13 undergo nuclear envelope breakdown, and mature stage 14 oocytes are arrested at metaphase I. Egg activation is triggered by rehydration and mechanical stimulation during ovulation, just prior to and independently of fertilization (*Horner and Wolfner, 2008b*; *Krauchunas et al., 2013*). In many species, $Ca^{2+}$ signaling is involved in egg activation (*Stricker, 1999*); this is true also in Drosophila despite the lack of a role for sperm entry (*Kaneuchi et al., 2015*). When activated oocytes are laid without fertilization (unfertilized activated eggs), they are arrested after the completion of meiosis. Although maternal mRNAs are translated, mitosis does not begin (*Doane, 1960*; *Mahowald et al., 1983*; *Kronja et al., 2014b*). Fertilized embryos enter into rapid embryonic mitotic cycles, which consist of alternating S- and M-phases.

Our recent genome-wide study has shown that hundreds of maternal mRNAs are translationally up- or down-regulated during the oocyte-to-embryo transition and that the majority of these translational changes require PNG kinase (*Kronja et al., 2014b*; *Eichhorn et al., 2016*). *png* was identified originally as a gene necessary for initiation of mitosis in the embryo (*Shamanski and Orr-Weaver, 1991*). This requirement was shown to reflect *png* upregulating *cyclin B* (*cycB*) translation during the oocyte-to-embryo transition (*Vardy and Orr-Weaver, 2007*). Additionally, *png* was found to promote translation of *smg* mRNA, thus controlling degradation of maternal mRNAs later in embryogenesis (*Tadros et al., 2007*). *png* encodes a Ser/Thr kinase that has two regulatory subunits, PLU and GNU (*Freeman et al., 1986*; *Fenger et al., 2000*; *Lee et al., 2003*). Our biochemical analysis using purified recombinant proteins demonstrated that PNG forms a complex with PLU and GNU and that GNU dimerizes the PNG-PLU subcomplex to elevate its kinase activity (*Lee et al., 2003*). However, how PNG kinase activity is developmentally regulated during the oocyte-to-embryo transition remains unknown. Unraveling this mechanism is important for understanding coupling of the events induced by egg activation. Here we show that CDK1 phosphorylation of GNU negatively regulates PNG kinase activity. PNG kinase becomes activated after meiosis completion promoted by CDK1 inactivation and then self inactivates, limiting its activity to the transition between oocyte and embryo and providing the mechanism to coordinate completion of meiosis with regulation of maternal mRNA translation.

## Results

### Dynamics of PNG kinase complex components during oocyte maturation

To decipher the developmental control of PNG kinase, we first examined the dynamics of protein levels of the PNG complex subunits during the developmental window that includes oocyte maturation (egg chamber stages 10–14) (*Figure 1A,B*). PNG and PLU were observed in extracts from stage 10 oocytes, which are arrested in prophase I (*Figure 1A*). In contrast, GNU was undetectable in stage 10, but was present as maturation occurred in stage 13 (*Figure 1B*). The protein levels of all three subunits were high in mature oocytes (stage 14), which are arrested at metaphase I (*Figure 1A,B*), a conclusion supported by quantitative mass spectrometry experiments (*Kronja et al., 2014a*). These results show protein expression of all the PNG complex components during oocyte maturation, but GNU protein is expressed in a narrower time window during very late oogenesis. The accumulation of PNG complex components during oocyte maturation could require CDK1 activity, because in *twine* mutant stage 14 oocytes, PNG complex proteins did not accumulate (*Figure 1—figure supplement 1*). *twine* encodes a homolog of the Cdc25 phosphatase required to activate CycB/CDK1 during oogenesis (*Courtot et al., 1992*).

Next we investigated protein levels of the PNG kinase complex components after egg activation. Egg activation is triggered by ovulation, and activated oocytes rapidly resume and complete meiosis in the oviduct (*Heifetz et al., 2001*), preventing recovery of oocytes in intermediate meiotic stages after egg activation. A powerful in vitro egg activation system, however, provides a means to follow the events after egg activation (*Mahowald et al., 1983*; *Page and Orr-Weaver, 1997*; *Horner and Wolfner, 2008a*). In this system, isolated stage 14 oocytes are activated by hypotonic buffer treatment, and the in vitro activated oocytes reproduce the events normally occurring rapidly in the oviduct, including meiotic cycle progression and protein synthesis (*Page and Orr-Weaver, 1997*; *Horner and Wolfner, 2008a*). In the in vitro activated oocytes, the protein levels of PNG complex components did not show major changes except for a slight decrease in GNU levels (*Figure 1A,B*).

Notably, a change in GNU mobility occurred during egg activation. A slower mobility form of GNU present following oocyte maturation was replaced by a faster mobility form between 20 and 30 min after hypotonic buffer treatment. These changes are consistent with a loss of phosphorylation, and indeed the slower mobility forms of GNU were previously noted in oocytes and shown to be due to phosphorylation (*Renault et al., 2003*; *Zhang et al., 2004*). The faster mobility form of GNU likely is hypophosphorylated rather than completely dephosphorylated, as phosphoproteome studies showed that GNU is phosphorylated at several sites in embryos (*Zhai et al., 2008*). The upper bands of GNU disappeared when okadaic acid (OA), a PP2A phosphatase inhibitor, was omitted from the native lysis buffer, implying that the slower mobility is due to phosphorylation and

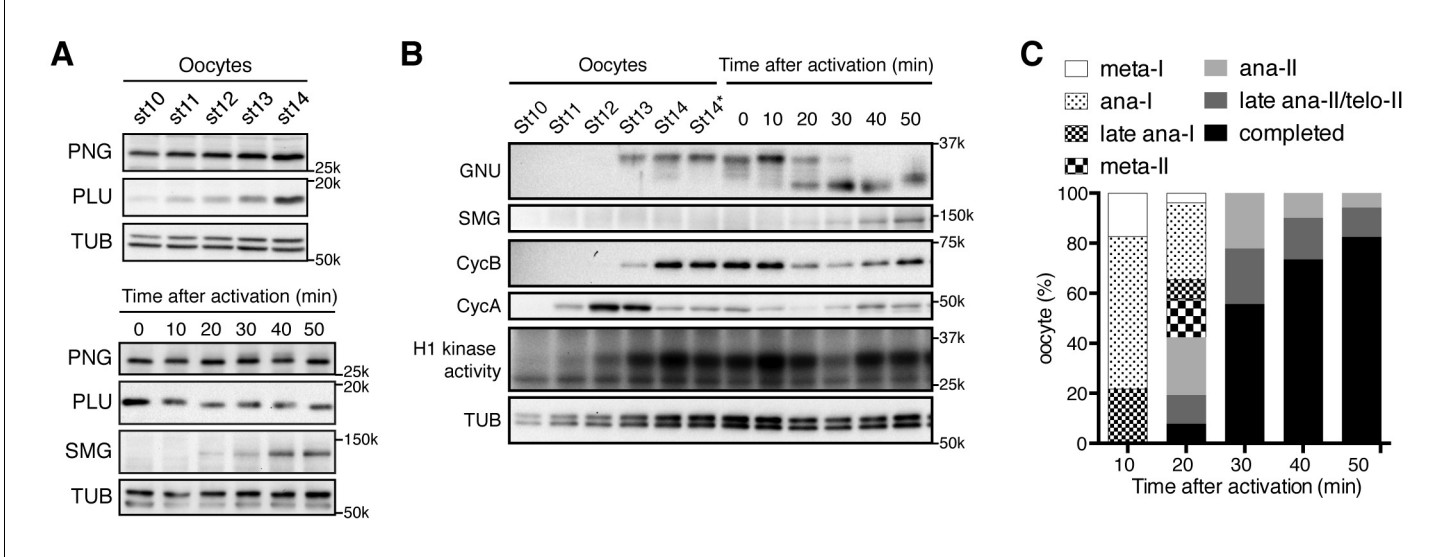

**Figure 1.** PAN GU (PNG) kinase activation coincides with GIANT NUCLEI (GNU) dephosphorylation after the completion of meiosis. (**A**) Dynamics of PNG and PLU proteins during oocyte maturation and after egg activation. Protein levels of PNG and PLU in oocytes in the indicated stages of oocyte development and in in vitro activated oocytes at the indicated times after hypotonic buffer treatment (activation) were examined by immunoblotting. Alpha Tubulin (TUB) was examined as a loading control. SMAUG (SMG) was examined as the indicator of PNG activation. (**B**) Dynamics of GNU and mitotic Cyclin proteins during oocyte maturation and after egg activation. GNU and the mitotic CycA and B in oocytes at the indicated stages of oocyte development and in in vitro activated oocytes at the indicated times after hypotonic buffer treatment (activation) were examined by immunoblotting. Total CDK1 activity was measured as H1 kinase activity. Alpha Tubulin (TUB) was examined as a loading control. SMG was examined as the indicator of PNG activation. Stage 14 oocytes before or after being held in the female and thus dehydrated (St14, St14*) were examined (*Mahowald et al., 1979*), because held stage 14 oocytes were used for in vitro egg activation (0 min; see Materials and methods). (**C**) Meiotic progression in in vitro activated oocytes. In vitro activated oocytes were fixed at specific time points after hypotonic buffer treatment (activation), the DNA stained with DAPI, and the meiotic stage in the activated oocytes was determined by DNA morphology and quantified. All of the results shown were repeated in two biological replicates except the PLU immunoblot in the bottom of panel 1A, which was done one time.

The following source data and figure supplements are available for figure 1:

**Source data 1.** Timing of each stage of meiosis following egg activation.

**Figure supplement 1.** *twine* is required for accumulation of PNG kinase complex component proteins in oocytes.

**Figure supplement 2.** GNU mobility shift is sensitive to phosphatase activity.

**Figure supplement 3.** The hyperphosphorylated form of GNU disappears at the end of meiosis II.

sensitive to phosphatase (*Figure 1—figure supplement 2*). We observed that GNU protein in oocytes is easily dephosphorylated during preparation of oocyte lysates depending on how rapidly the lysates were prepared and whether they were diluted quickly. Thus in some of our early experiments lysates were prepared less rapidly, causing both the phosphorylated and hypophosphorylated forms to be observed (*Figure 1—figure supplement 1*).

Given the dephosphorylation of GNU that accompanies egg activation, we wanted to test when this occurred relative to the completion of meiosis. Lysates were prepared from eggs at time points after activation and monitored for CDK activity by histone H1 phosphorylation or levels of the mitotic Cyclins A and B (CycA and B) (*Figure 1B*). As observed previously (*Vardy et al., 2009*), CycA levels increased and then declined during oocyte maturation. CycB was accumulated and its levels decreased 20 min after egg activation, beginning to increase again at 40 min. The levels of CycB correlate with oscillation of H1 kinase activity. Activated eggs from each time point were fixed, stained with DAPI, and examined for meiotic stage (*Figure 1C*, *Figure 1—source data 1*). The 20 min post activation time point marks the point when the majority of eggs were completing meiosis, and by 30 min nearly 60% had completed meiosis. In addition to these population studies in which there could

be some variability, individual activated eggs were examined for meiotic stage, and then lysates prepared from each egg and examined for the GNU form on western blots (*Figure 1—figure supplement 3*). The transition from the fully phosphorylated to the hypophosphorylated form was observed to occur between early and late anaphase of meiosis II (anaphase II). Thus we conclude GNU becomes dephosphorylated when CycB levels decline, H1 kinase activity decreases, and meiosis is completed.

## PNG kinase is activated during meiosis II completion, concurrent with GNU dephosphorylation

The presence of all three PNG kinase subunits in mature oocytes, despite the effects of the kinase on translation not being manifest until after egg activation, posed the question of when and how the kinase is activated. The change in GNU form raised the possibility that dephosphorylation of GNU might be linked to PNG activation. To define the timing of PNG activation, we used SMG protein accumulation as an indicator of PNG activity after attempts to immunoprecipitate PNG kinase from activated eggs to measure kinase activity directly failed to obtain sufficient amounts of PNG complex for the assay. Because *smg* maternal mRNA translational activation is a downstream target dependent on PNG, appearance of SMG protein after egg activation requires PNG activity (*Tadros et al., 2007*; *Kronja et al., 2014b*; *Eichhorn et al., 2016*). As detailed above, SMG protein levels were examined by western blots of extracts from staged activated eggs. Strikingly, we found that SMG started to accumulate 20 to 30 min after hypotonic buffer treatment, when GNU is dephosphorylated and meiosis completed (*Figure 1A,B*). Thus we hypothesized that phosphorylated GNU is unable to activate PNG kinase, and that dephosphorylation at the completion of meiosis leads to PNG activation.

## In mature oocytes GNU is phosphorylated at CycB/CDK1 sites

To determine which residues in GNU were phosphorylated in mature oocytes, we generated strains expressing a functional *gnu-gfp* transgene (see Figure 3) under the normal regulatory elements for *gnu*. We isolated stage 14 oocytes, immunoprecipitated the GNU-GFP fusion protein, and identified phosphosites by mass spectrometry (*Figure 2A*, *Figure 2—source data 1*). There are nine predicted CycB/CDK1 sites (S/T-P) in GNU, and peptides containing these residues were the phosphopeptides most frequently recovered (*Figure 2—figure supplement 1*). All nine sites were detected as phosphorylated, with significant Ascore values (*Figure 2—figure supplement 1*). Eight of these sites were recovered both in trypsin and chymotrypsin digests and had Ascore values of greater than 95% significance, whereas T158 was identified only from a trypsin digest and scored 90% significant. In addition to the CycB/CDK1 sites, four other phosphosites were recovered with low numbers of phosphopeptides, although the Ascore values were significant. Eight of the nine CycB/CDK1 sites are in a region in the GNU protein predicted to be disordered by four different algorithms (*Figure 2—figure supplement 2*). This is of interest because phosphorylation of disordered protein regions has been observed to affect protein-protein interactions (*Holt et al., 2009*).

The identification of CycB/CDK1 phosphosites on GNU in mature oocytes is consistent with the timing of the GNU phosphorylation change, high at metaphase I and decreased by late anaphase II. CycB/CDK1 activity governs M-phase progression (*Kishimoto, 2015*). The decline in H1 kinase and CycB protein levels coincident with dephosphorylation of GNU also were consistent with its phosphorylation being dependent on CycB/CDK1. We tested potential roles for two other kinases. GNU is phosphorylated by PNG in vitro and shows a mobility shift on SDS-PAGE (*Lee et al., 2005*). Mature oocytes from *png* mutant mothers (*png*[1058]/ *png*[1058]), however, still have the hyperphosphorylated form of GNU, suggesting that PNG activity is not required for the mobility shift (*Figure 2—figure supplement 3*). This also is consistent with the results that PNG kinase becomes activated after egg activation. We noted that MAPK, which is active during oocyte maturation (*Ivanovska et al., 2004*; *Sackton et al., 2007*) and phosphorylates S/T-P, seems to be dispensable for the slowest mobility GNU phosphorylated form. In null mutants of *mos*, the female-meiosis specific MAPK kinase kinase, phosphorylated active MAPK is decreased to less than 10% of wild-type levels, but the mobility of GNU was identical to heterozygous *mos* sibling controls (*Figure 2—figure supplement 3*). Thus we focused on the biochemical consequences of phosphorylation of GNU by CycB/CDK1, evaluating all nine sites detected by mass spectrometry.

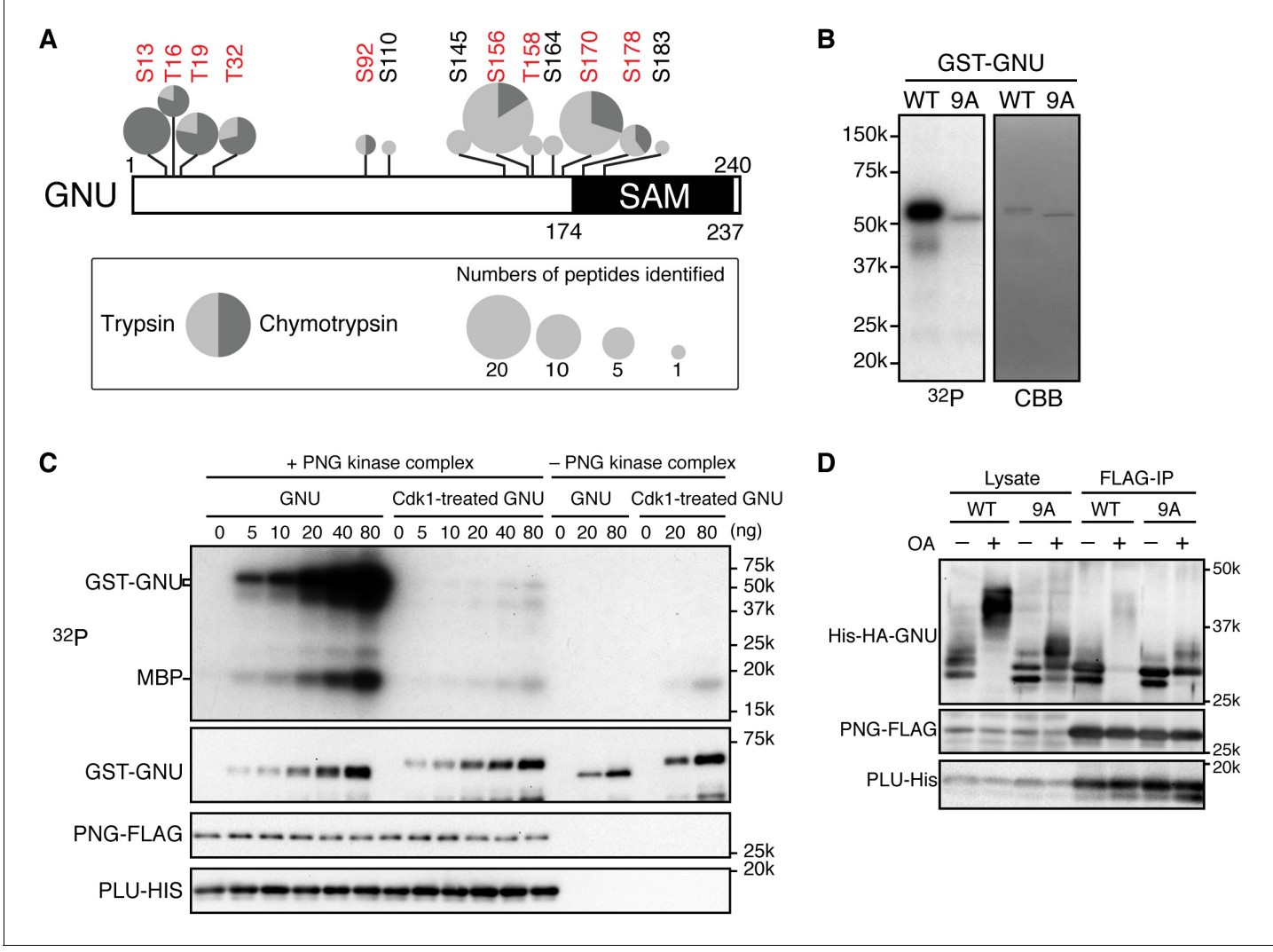

**Figure 2.** Cyclin B (CycB)/CDK1 phosphorylates GNU and in vitro inhibits its ability to activate PNG kinase. (A) A schematic representation of GNU protein and the phosphorylated residues identified in st14 oocytes. Overall coverage was 88%. In total, 93 phosphopeptides from tryptic and chymotryptic digests were assigned with >90% certainty based on Ascore. 62 out of the 93 phosphopeptides were with >99% certainty (see also supplemental figure). The region between amino acids 116 and 185 was poorly covered in the chymotrypsin analysis (supplement MS data). Phosphorylation sites in red match the S/T-P motif and were considered in further experiments. The size of the pie charts shows numbers of phosphopeptides assigned with >90% certainty. The light and dark gray indicate tryptic and chymotryptic peptides, respectively. The sterile alpha motif (SAM) in the C terminus of the protein is shown. (B) CDK1 directly phosphorylates GNU. GST-GNU WT or a mutant with alanine substitution of the nine S/T-P sites (9A) shown in red in (A) were incubated with purified active CycB/CDK1 in the presence of radioactive ATP. Phosphorylation of GST-GNU WT or 9A was detected by autoradiography (left panel, $^{32}$P). GST-GNU protein levels were examined by coomassie staining (right panel, CBB). This experiment was done twice. (C) CDK1 phosphorylation inhibits the ability of GNU to activate the PNG kinase. PNG kinase activity was measured by phosphorylation of myelin basic protein (MBP) (upper panel, $^{32}$P). GST-GNU treated or untreated with CycB/CDK1 was added into the kinase reaction in the nanogram (ng) amounts indicated. Note that GNU is also phosphorylated by PNG kinase. GST-GNU was incubated with MBP in the absence of PNG kinase complex as a negative control. The protein levels of the PNG kinase complex components in the reaction were examined by immunoblotting, confirming the phosphorylation state of GNU (bottom panels). (D) Phosphorylation at CDK1 sites in GNU prevents the interaction between GNU and the PNG-PLU subcomplex. Wild-type GNU binding to the PNG-PLU subcomplex was inhibited in okadaic acid (OA) -treated Sf9 cells, but alanine substitution in the nine S/T-P sites (9A) of GNU restores the binding. PNG-FLAG and PLU-His were expressed in Sf9 insect cells with His-HA-GNU WT or 9A (Lysate). The PNG complex expressing cells were incubated with or without OA, a PP2A inhibitor. PNG kinase complex formation was examined by immunoprecipitation of PNG-FLAG (FLAG-IP) followed by immunoblotting. Three biological replicates of this experiment were done.

The following source data and figure supplements are available for figure 2:

**Source data 1.** Phosphopeptides recovered by mass spec and their mass spec properties.

*Figure 2 continued on next page*

*Figure 2 continued*

**Figure supplement 1.** MS/MS spectra resulting from HCD fragmentation of phosphopeptides containing GNU.

**Figure supplement 2.** CDK1 phosphorylation sites of GNU are in a predicted disordered region.

**Figure supplement 3.** PNG and MAPK activity are dispensable for GNU hyperphosphorylation in mature oocytes.

**Figure supplement 4.** Phosphorylation at CDK1 sites in GNU blocks PNG kinase complex formation in Sf9 cells.

## CycB/CDK1 phosphorylates GNU and inhibits its ability to activate PNG kinase in vitro

To evaluate the functional consequences of GNU phosphorylation in mature oocytes, we first tested whether CycB/CDK1 directly phosphorylates GNU in vitro. We found that purified active CycB/CDK1 directly phosphorylated recombinant GNU (*Figure 2B*), resulting in its mobility shift-up on SDS-PAGE. When we introduced mutations that substituted alanine into all nine CDK1 sites of GNU (GNU-9A), GNU phosphorylation and mobility shift-up by CycB/CDK1 were blocked (*Figure 2B*), indicating that GNU indeed is a direct substrate of CycB/CDK1 in vitro.

To examine the consequences of GNU phosphorylation by CycB/CDK1 on PNG kinase activation, we assayed PNG kinase activity in vitro, using myelin basic protein (MBP) as an in vitro substrate. Purified recombinant PNG kinase complex was activated by adding recombinant GNU into the kinase reaction, and the level of activation responded in a dose-dependent manner to GNU addition (*Figure 2C*). GNU added to the reaction also was phosphorylated by PNG, consistent with GNU being a known in vitro substrate of PNG (*Lee et al., 2005*). Strikingly, GNU that was phosphorylated by CycB/CDK1 prior to addition to the kinase assay did not elevate PNG kinase activity (*Figure 2C*). This showed that CycB/CDK1-phosphorylation of GNU blocks the ability of GNU to activate PNG kinase.

## Phosphorylation of CDK1 sites in GNU interferes with its interaction with the PNG-PLU sub-complex

Because GNU has been demonstrated to form a complex with the PNG-PLU sub-complex to activate PNG kinase (*Lee et al., 2003*), next we investigated whether phosphorylation of GNU by CycB/CDK1 would affect complex formation. We expressed PNG, PLU and GNU in Sf9 cells and analyzed complex formation by PNG immunoprecipitation. As previously shown, PNG, PLU and GNU co-immunoprecipitate, indicating their physical association as a complex in Sf9 cells (*Figure 2D*) (*Lee et al., 2003*). In contrast, when the cells expressing the PNG kinase complex were treated with OA, which causes an M-phase-like state (*Yamashita et al., 1990*), GNU was hyper-phosphorylated and dissociated from the PNG-PLU sub-complex (*Figure 2D*). Remarkably, introducing alanine mutations into the CDK1 consensus sequences suppressed the hyperphosphorylation of GNU and restored interaction of GNU with the PNG-PLU sub-complex in the OA-treated cells (*Figure 2D*, *Figure 2—figure supplement 4*). These results indicate that phosphorylation of GNU on CDK1 target sites prevents its interaction with the PNG-PLU sub-complex. Together with the results of the in vitro kinase assays (*Figure 2C*), this leads to the conclusion that CDK1 inhibits PNG kinase activity via preventing assembly of the full PNG kinase complex, which is essential for its kinase activation (*Lee et al., 2003*).

## GNU-9A activates PNG kinase prematurely in stage 14 oocytes, affecting meiotic progression after egg activation

The above results prompted us to examine the consequences of replacing GNU in oocytes with the GNU-9A form that is not phosphorylated by CycB/CDK1, the prediction being that this would lead to premature complex formation and activation of PNG kinase. We generated transgenic lines in which the *gnu-9A* gene was expressed under its normal regulatory elements and crossed these into *gnu* protein null mutants. The ability of GNU-9A to replace wild-type GNU was evaluated by

examining the phenotype of embryos laid by mothers transgenic for a control wild-type *gnu-gfp* (*gnu-gfp WT*) transgene to those bearing the *gnu-gfp 9A* transgene (*Figure 3*, *Figure 3—source data 1*). Two different transgene insertion sites were examined for each of the *gnu-9A* (2–7 and 2–8) and wild-type *gnu* (1–5 and 1–8) transgenes. In the complete absence of PNG activity meiosis is completed, but due to decreased levels of CycB protein the first mitosis of embryogenesis does not occur (*Lee et al., 2001*; *Vardy and Orr-Weaver, 2007*). This results in laid eggs with a single, multi-lobed, polyploid nucleus or up to five polyploid nuclei corresponding to the four meiotic products and the male pronucleus (*Figure 3A*). With partial PNG activity several mitotic divisions occur, but then mitosis ceases and these become polyploid (*Fenger et al., 2000*). The GNU-GFP WT fusion protein complemented the *gnu* mutant to produce embryos in which normal embryonic mitoses occurred (*Figure 3A*). GNU-GFP 9A had partial function: all embryos underwent mitosis, but some

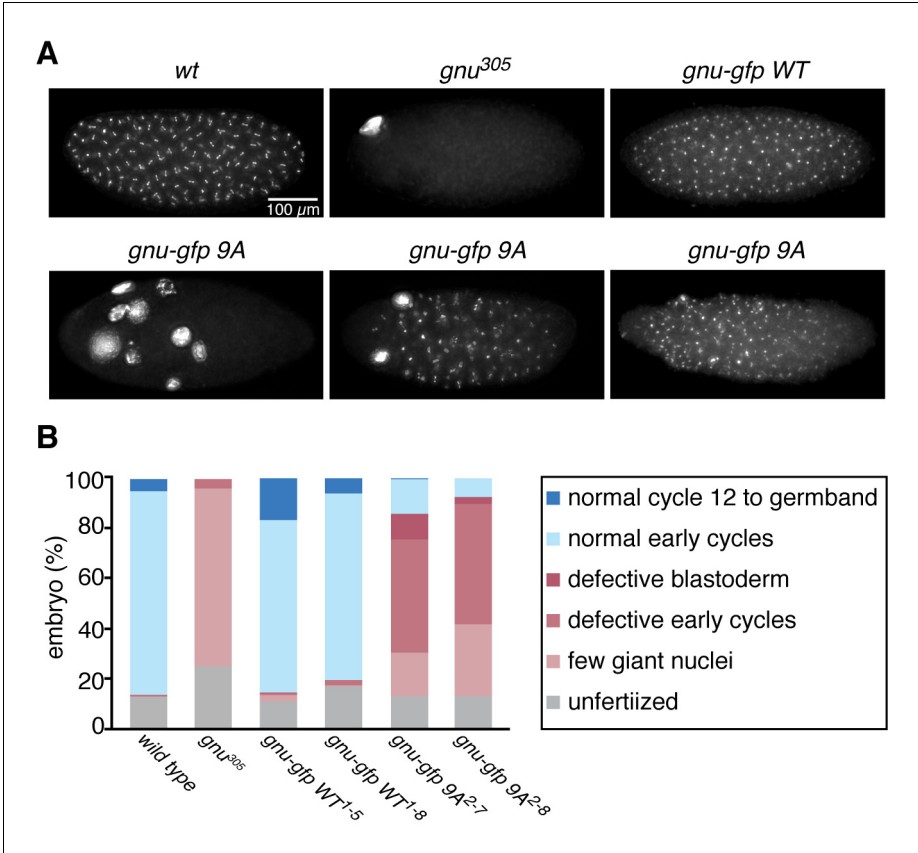

**Figure 3.** The GNU-9A protein can partially rescue *gnu*[305]/*gnu*[305] mutants. (A,B) Fertilized embryos were collected for 2 hr from females of genotype *wt*, *gnu*[305]/*gnu*[305] (*gnu*[305]), *gnu*[305]/*gnu*[305];[*gnu-gfp WT*] (*gnu-gfp WT*) or *gnu*[305]/*gnu*[305];[*gnu-gfp 9A*] (*gnu-gfp 9A*), fixed and stained with DAPI. The embryos from the *wt* and *gnu-gfp WT* females show normal early cycles, and the *gnu*[305] embryo is classified as having a few giant nuclei. Several phenotypes are shown for *gnu-gfp 9A*, from left to right: few giant nuclei, defective early cycles, and defective blastoderm. (B) Quantification of the embryo phenotypes. Two independent lines for each transgene were tested (*gnu-gfp WT*[1-5] and *WT*[1-8] and *9A*[2-7] and *9A*[2-8]). At least 300 embryos were scored for each of the transgene lines, and 150 were scored for the wild-type and *gnu*[305] controls. The data are the sum of two technical replicates. A second biological replicate experiment was scored and comparable rescue results observed, but the embryos were not quantified.

The following source data and figure supplement are available for figure 3:

**Source data 1.** The numbers of embryos in each phenotypic class scored for each genotype.

**Figure supplement 1.** Alanine substitution in the nine S/T-P sites in GNU prevents its ability to activate PNG kinase in in vitro.

exhibited the phenotype of weak alleles of *png* complex subunits, whereas others were able to complete the syncytial mitotic divisions (*Figure 3B*). The *gnu-9A* 2–7 line appeared to have slightly more rescue activity than the 2–8 line, and this correlated with higher expression of the GNU-GFP 9A protein in the 2–7 line (*Figure 4A*). In vivo replacement of GNU with GNU-9A is the most sensitive test of activity. We also examined the ability of GNU-9A to activate PNG kinase in vitro, and we observed that its in vitro activity is extremely low (*Figure 3—figure supplement 1*).

Given that GNU-GFP 9A is partially functional, we were able to exploit it to investigate the consequences of the presence in mature oocytes of GNU that is unable to be phosphorylated by CycB/CDK1. If this resulted in developmentally premature activation of PNG kinase, we predicted to see elevated levels of CycB protein, because *cycB* mRNA is translated throughout late oogenesis and PNG enhances its translation at egg activation (*Eichhorn et al., 2016*). We found that levels of CycB protein in stage 14 oocytes were significantly higher when GNU-GFP 9A was expressed compared to GNU-GFP WT (*Figure 4A,B*, five replicates, p=0.03 for line 2–7 and p=0.01 for line 2–8, *Figure 4—source data 1*). This is consistent with hypophosphorylated GNU leading to prematurely active PNG in mature oocytes. As expected, in stage 14 oocytes, GNU-GFP 9A migrated faster than GNU-GFP WT, indicating that indeed the 9A mutant was not phosphorylated at CycB/CDK1 sites stage 14 oocytes (*Figure 5*). Strikingly, GNU-GFP 9A, but not GNU-GFP WT, was detected at low levels but consistently in PNG immunoprecipitates from stage 14 oocytes (*Figure 5*). Taken together, these results confirm that interaction between the PNG-PLU sub-complex and GNU is prevented by phosphorylation of CDK1 target sites on GNU. The phosphorylation-resistant form of GNU is able to associate with PNG and results in premature elevation of Cyc B protein levels, consistent with PNG kinase being prematurely active.

The apparent premature activation of PNG in mature oocytes did not affect the metaphase I configuration of chromosomes in mature oocytes. We used in vitro activation to examine whether the timing or accuracy of meiotic divisions was altered by the increased levels of CycB resulting from inhibition of GNU phosphorylation by CycB/CDK1. At egg activation the chromosomes at metaphase I condense and become individually distinguishable (*Figure 4D*) (*Page and Orr-Weaver, 1997*). In the GNU-GFP 9A oocytes the condensed phase of metaphase I persisted longer than in wild type, and the metaphase I/anaphase I transition was delayed (*Figure 4C*, *Figure 4—source data 1*). These results were consistent with increased levels of CycB taking longer to be targeted for degradation mediated by the APC/C E3 ubiquitin ligase to trigger the metaphase I/anaphase I transition. After this transition, however, the levels of Cyclin B declined somewhat more rapidly in the GNU-GFP 9A compared to wild type. This may reflect the reduced activity of GNU-9A compared to wild type and the lower levels of GNU-GFP 9A protein in the 2–8 transgenic line; thus despite premature PNG activation and elevated *cycB* mRNA translation in mature oocytes, after the completion of meiosis there are reduced levels of active PNG kinase to promote full *cycB* translation relative to wild type. This would account for the limited number of activated eggs with GNU-9A that form polar bodies, the condensed meiotic products whose formation is dependent on reactivation of CycB/CDK1 (*Vardy and Orr-Weaver, 2007*).

In addition to delays in progression through meiosis, the GNU-GFP 9A form resulted in defects in meiotic chromosome segregation. The morphology of chromosomes after egg activation at metaphase I was aberrant in 19–23% of the oocytes. The chromosomes either were delayed in congressing to the metaphase I plate or had incorrect alignment on the plate. Later, as meiosis was completed, the size and DAPI-staining intensity was uneven in up to 23% of oocytes, reflecting unequal segregation of chromosomes during the meiotic divisions (*Figure 4D,E*, *Figure 4—source data 1*). Because there are only four chromosomes in Drosophila and the two large autosomes are the same size, visualization of aberrant chromosome alignment and segregation underestimates the rate of meiotic errors. Thus we conclude that proper levels of CycB protein and possibly other PNG targets are necessary for the accuracy of meiotic divisions in the oocyte. The simplest interpretation of the effects of GNU-GFP 9A in oocytes is that the dephosphorylated form of GNU leads to premature PNG activation.

## Inactivation of PNG following egg activation

We previously observed that GNU protein levels were decreased in in vivo activated eggs and early embryos (0–2 hr) (*Kronja et al., 2014b*). PNG and PLU protein levels decreased, but much later in embryogenesis (*Elfring et al., 1997*; *Fenger et al., 2000*). Here we found that the decrease of GNU

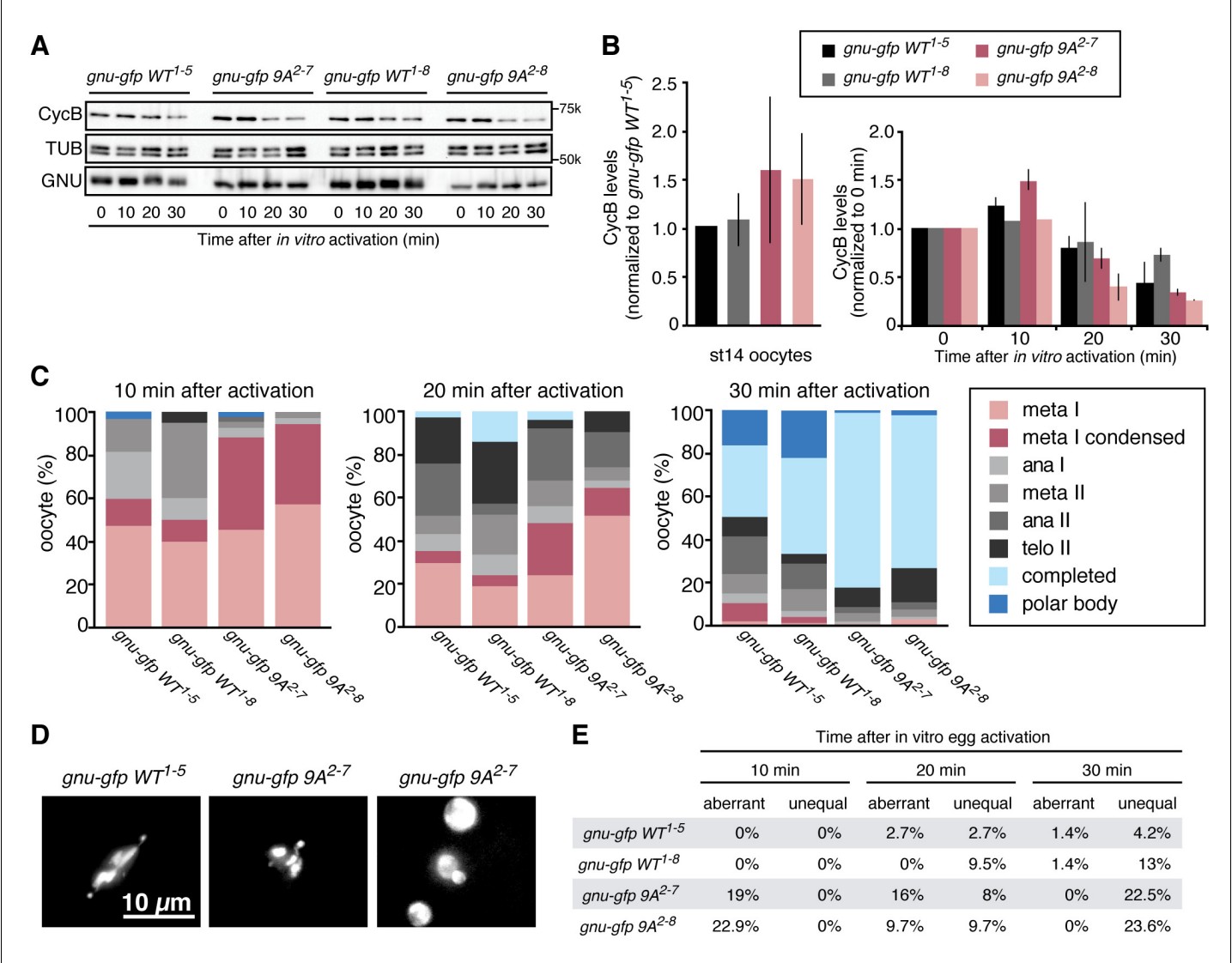

**Figure 4.** Phosphorylation-resistant GNU increases CycB protein levels in mature oocytes and affects meiotic progression after egg activation. (**A,B**) Stage 14 oocytes (0 min) were isolated from *gnu305*/*gnu305* females expressing *gnu-gfp WT* or *9A* (*gnu-gfp WT* or *gnu-gfp 9A*) transgenes and were activated in vitro by hypotonic buffer treatment. CycB, TUB and GNU-GFP (GNU) protein levels were examined by immunoblotting at the indicated times after hypotonic buffer treatment (activation). The two independent lines for each transgene were tested (*gnu-gfp WT1-5* and *WT1-8* and *9A2-7* and *9A2-8*). (**B**) CycB levels in (**A**) were quantified and normalized with TUB levels. The left graph in (**B**) shows the CycB levels in each line relative to the levels of gnu-gfp *WT1-5*. The error bars correspond to five biological replicates. The CycB protein levels are significantly higher in the *gnu-gfp-9A* oocytes from both transgene lines compared to *gnu-gfp WT* oocytes by a T test. The right graph in (**B**) shows the dynamics of CycB levels after egg activation in each line relative to their levels at 0 min. These data are from immunoblots from two biological replicates, except for the 10 min time point for lines 1–8 and 2–8, which were done once. (**C**) Meiotic progression is delayed in *gnu-gfp-9A* oocytes compared to *gnu-gfp WT* oocytes. Stage 14 oocytes (0 min) were isolated from *gnu-gfp WT* or *9A* females and were activated in vitro by hypotonic buffer treatment. The activated oocytes were fixed at indicated times after hypotonic buffer treatment (activation) and were stained with DAPI to define their meiotic stages. The data for the 10 min time point are compiled from three biological replicates. The 20 and 30 min time points were done once. At least 30 but up to 140 oocytes were scored for each genotype for each data point. (**D**) Phospho-resistant GNU causes defects in chromosome alignment and segregation after egg activation. Chromosomes and post-meiotic nuclear morphology were examined in activated oocytes expressing the indicated transgene in a *gnu305* mutant background. Representative images show a normal meiosis I (left: *gnu-gfp WT1-5*), aberrant meiosis I (middle: *gnu-gfp 9A2-7*) and interphase nuclei resulting from unequal segregation during meiosis (right *gnu-gfp 9A2-7*). (**E**) The percentage of oocytes with aberrant chromosome morphology or unequal chromosome segregation was scored at each time point for the indicated genotypes. The quantification is from one biological replicate. At least 20 but up to 140 oocytes were scored for each data point.

The following source data is available for figure 4:

*Figure 4 continued on next page*

*Figure 4 continued*

**Source data 1.** Quantification of Cyclin B protein levels, the number of eggs in each meiotic stage following egg activation, and the number of eggs with aberrant or unequal meiotic divisions.

protein requires PNG kinase activity. In contrast to control embryos (*png*$^{1058}$/+) in which GNU protein levels were largely decreased, its levels remained high in *png* mutant fertilized embryos (*png*$^{1058}$/ *png*$^{1058}$), despite its becoming dephosphorylated (*Figure 6*). The dependency of GNU reduction on functional PNG kinase provides a mechanism to terminate PNG kinase function immediately after egg activation, restricting its activity to the narrow developmental window of the oocyte-to-embryo transition.

## Discussion

The massive changes in mRNA translation accompanying egg activation occur in a matter of minutes and must be linked to completion of meiosis in the oocyte. Here we find that in Drosophila the solution to this developmental challenge is the regulation of PNG kinase activity (*Figure 7*). Our results show that GNU is phosphorylated at CycB/CDK1 sites in mature oocytes, and in vitro CycB/CDK1 can directly phosphorylate GNU and thereby inhibit its ability to activate PNG kinase via inhibition of formation of the complex. In mature oocytes that are arrested at metaphase I, GNU is phosphorylated at CDK1 consensus sites and prevented from interaction with the PNG-PLU sub-complex. Following egg activation, as CycB protein and H1 kinase activity decline, GNU is dephosphorylated. This corresponds to the completion of meiosis and activation of the PNG kinase complex. Consistent with dephosphorylation of GNU being the crucial event for activation of PNG, substitution of a phosphorylation-resistant form of GNU into the oocyte results in premature elevation of CycB protein, implying PNG activation. Thus we propose that control of PNG kinase activity via GNU phosphorylation by CycB/CDK1 links meiotic completion and translational control of maternal mRNA to coordinate their timing precisely during egg activation (*Figure 7*). Active PNG leads to decreased GNU protein levels. This makes a negative feedback to shut down PNG kinase activity, thereby ensuring PNG kinase activity is constrained to the short developmental window of the oocyte-to-embryo transition (*Figure 7*).

Our findings highlight the linchpin role the GNU subunit plays in the developmental control of PNG kinase activity. This regulation is exerted both at the levels of GNU protein and via its phosphorylation state. Although the presence of all three PNG kinase complex proteins is limited to late oogenesis through the early embryo, GNU is present over a narrower time window. GNU protein is undetectable in stage 10 oocytes and is rapidly accumulated during oocyte maturation. Our previous genome-wide study showed that GNU protein accumulation during oocytes maturation relies on translational activation of its mRNA (*Kronja et al., 2014a*). Although the regulatory mechanisms for *gnu* translational activation remain to be defined, it is clearly dependent on CDK1, but not MOS, activity. Thus CDK1 promotes the appearance of GNU, permitting all PNG subunits to be present in the mature oocyte and poised for activation, while it simultaneously prevents activation by phosphorylation of GNU.

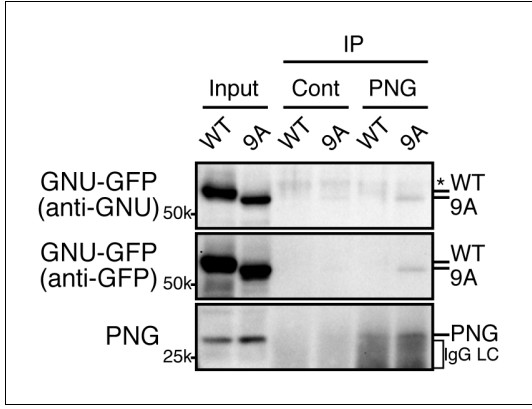

**Figure 5.** The GNU-9A mutant forms a complex with PNG prematurely in mature oocytes despite active CycB/CDK1. The GNU-9A mutant, but not wild-type GNU, associates with PNG in meta-I arrested mature oocytes. PNG was immunoprecipitated from extracts of stage 14 oocytes, which expressed GNU-GFP WT or the 9A mutant. The immunoprecipitates with PNG antibody were examined by immunoblotting. Asterisk indicates a nonspecific signal. IgG LC shows bands of the IgG antibody light chain. This experiment was repeated in two biological replicates.

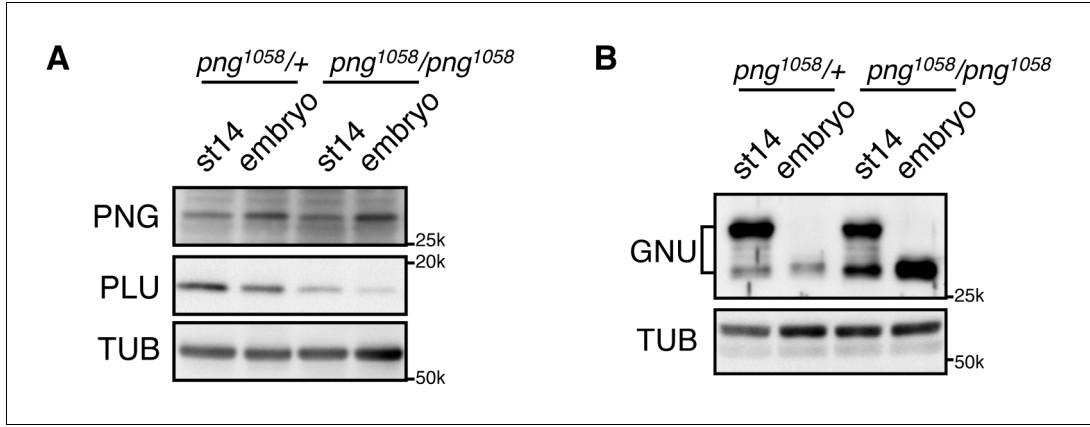

**Figure 6.** GNU protein levels decrease in embryos in a PNG-dependent manner. (**A**) PNG, and PLU levels in stage 14 oocytes (st14) or 0–2 hr collections of fertilized embryos (embryo) from $png^{1058}/FM6$ ($png^{1058}/+$) or $png^{1058}/png^{1058}$ females were examined by immunoblotting. Alpha Tubulin (TUB) was used as a loading control. PNG and PLU levels were hardly changed during the first two hours of embryo development. Note that PLU levels are reduced in *png* mutant oocytes. (**B**) GNU levels in stage 14 oocytes (st14) or 0–2 hr collections of fertilized embryos (embryo) from $png^{1058}/+$ or $png^{1058}/png^{1058}$ females were examined by immunoblotting. TUB was used as a loading control. GNU becomes dephosphorylated after egg activation in both the control and mutant, but is stabilized in the absence of functional *png*. This experiment was repeated in two biological replicates.

Several previous experimental observations now have clear significance in support of the role of CDK1 phosphorylation of GNU in inhibiting PNG complex formation and kinase activation. Ectopic GNU protein expression in early stage oocytes, which are in prophase I, causes *png*-dependent premature CycB protein expression and actin disorganization (*Renault et al., 2003*; *Vardy and Orr-Weaver, 2007*). Importantly, these defects result from expression of GNU at a developmental stage (stage 11) when although PNG and PLU are present at low levels, CycB/CDK1 is not active, and thus would not be able to block formation of the PNG complex. Analysis of the phosphorylation state of GNU in mutants for the calcipressin Sarah (*sra*) or the meiosis-specific APC/C activator Cortex (*cort*) showed that GNU remains hyperphosphorylated in eggs laid from both types of mutant mothers (*Krauchunas et al., 2013*). Interestingly, both of these mutants fail to complete meiosis, with *sra* mutant eggs arrested in anaphase I, and *cort* mutants arrested in metaphase II (*Page and Orr-Weaver, 1996*; *Horner et al., 2006*; *Takeo et al., 2006*). The failure of GNU to be dephosphorylated in both of these mutants agrees with our demonstration that GNU becomes hypophosphorylated after the completion of meiosis.

Although PNG and PLU levels in embryos are gradually decreased 2–4 hr after laying (*Elfring et al., 1997*; *Fenger et al., 2000*) GNU seems to disappear after the completion of meiosis but before the initiation of embryogenesis. This is evidenced by GNU protein levels being decreased in unfertilized in vivo activated eggs, which complete meiosis II but do not enter into the embryonic cycles. This rapid GNU disappearance requires PNG kinase activity, revealing the existence of a negative feedback to shut off PNG activity shortly after its activation. It remains to be determined how GNU protein levels decline. PNG phosphorylates GNU on sites other than the CDK1 sites (Hara and Orr-Weaver, unpublished), and this phosphorylation may target GNU for degradation mediated by an E3 ubiquitin ligase of the SCF or the APC/C families. We noted that GNU levels fail to decline in *cort* mutants (*Kronja et al., 2014a*), which lack one form of the APC/C. This may be an indirect effect of failure of dephosphorylation of the CDK1 sites in GNU and thus absence of PNG activation, but it is also possible that GNU is targeted directly to the APC/C^cort after completion of meiosis.

A key question raised by the demonstration of the regulatory role of GNU phosphorylation is whether developmental control of phosphatase activity is required. One possibility is that the phosphatase responsible for dephosphorylation of the CDK1 sites in GNU is constitutively active. CycB/CDK1 activity is high in the metaphase I-arrested mature oocyte, but egg activation triggers meiotic resumption and CDK1 inactivation. Reduction of CycB/CDK1 activity in the presence of active

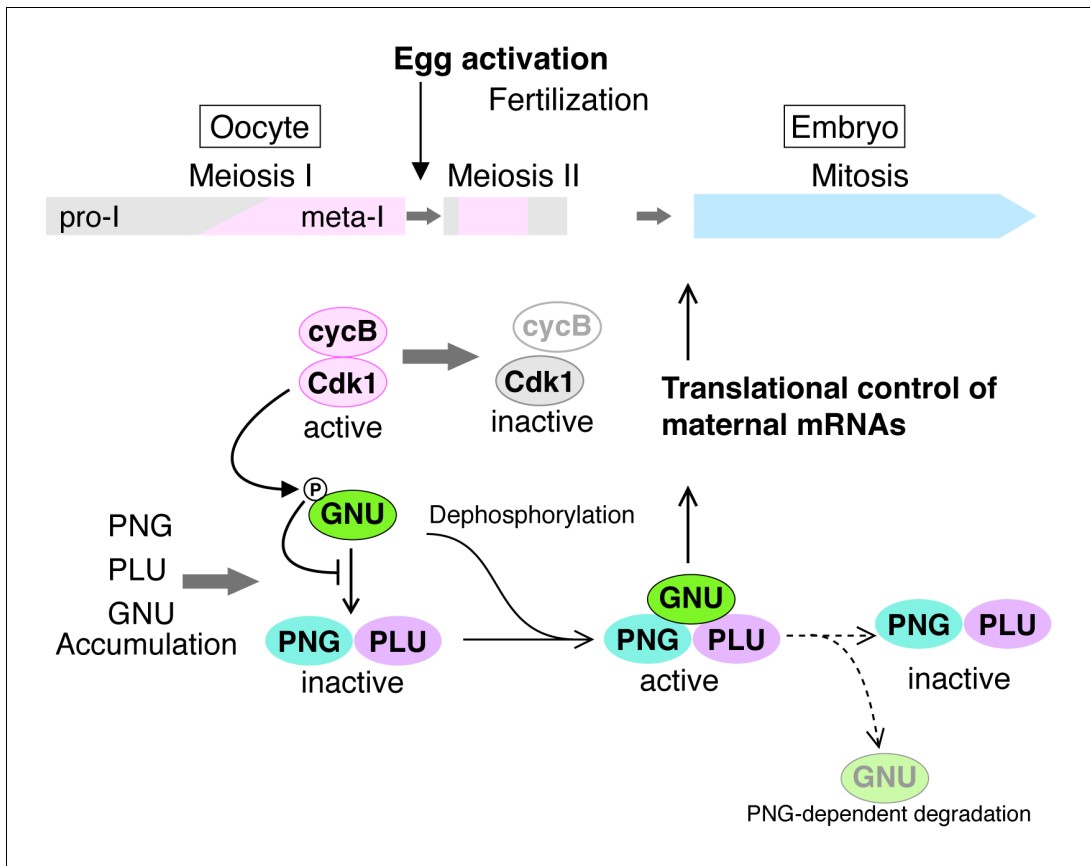

**Figure 7.** A model for developmental regulation of PNG kinase activity during the oocyte-to-embryo transition. Although mature stage 14 oocytes, which are arrested at metaphase of meiosis I, have the PNG kinase complex components, CycB/ CDK1 phosphorylates and inhibits GNU, preventing binding to PNG-PLU and active PNG kinase. Egg activation triggers resumption and completion of meiosis, which result in CDK1 inactivation. Subsequently, GNU becomes dephosphorylated, binds to PNG-PLU and activates PNG kinase, leading to translational control of maternal mRNAs, essential for initiation and progression of the mitotic cycle in embryos. Thus, phosphorylation of GNU by CDK1 couples cell cycle progression and maternal mRNA translation during the oocyte-to-embryo transition. PNG kinase activity also reduces GNU protein levels after egg activation. Together with the PNG activation mechanism, the negative feedback causing GNU protein reduction limits PNG kinase activation to this short time window, important for the precise oocyte-to-embryo transition.

phosphatase may be sufficient for hypophosphorylated GNU. Alternatively, egg activation could lead to activation of a phosphatase. The identity of the GNU phosphatase for the CDK1 sites awaits elucidation. Although it has been shown that PP1 is capable of dephosphorylating GNU in mature oocytes (*Renault et al., 2003*), PP2A also is a possible GNU phosphatase during the process, particularly given it is the major phosphatase that removes phosphates on CDK1 substrates in mitotic exit (*Mochida et al., 2009*).

The partial activity of the GNU 9A protein form provides insights into the mechanism by which GNU activates the PNG kinase. This mutant protein binds to the PNG-PLU sub-complex even in high CDK activity, however, it does not activate PNG kinase to the full extent that wild-type GNU does. This implies that there is an additional step to PNG kinase activation after GNU and the PNG-PLU sub-complex interaction. It is likely that the kinase activation following complex formation involves the N-terminus of GNU. A point mutation in the GNU N-terminal region (changing Pro 17 to Leu) retains ability to bind the PNG-PLU sub-complex but not does not activate PNG kinase activity in vitro (Hara and Orr-Weaver, unpublished and *Figure 3—figure supplement 1*).

The PNG kinase is significant for the understanding of how a kinase can rapidly control translation of hundreds of mRNAs. In addition to these insights into mRNA translation, identifying and defining

the role of regulators involved in triggering the profound changes accompanying the oocyte-to-embryo transition is crucial for our understanding of the onset of development, with implications for human fertility. Here we have shown two forms of regulation of PNG kinase activity: one being regulation of protein expression of PNG kinase complex components and another being regulation of its activity. Strikingly, a cell cycle regulator, CDK1, controls both. This implies that CDK1 precisely regulates PNG kinase activity, a translational regulator, thus coordinating cell cycle progression and the translational landscape change during the oocyte-to-embryo transition. There are interesting parallels between our findings and those in *C. elegans*. In *C. elegans* another kinase, the MBK-2 member of the conserved DYRK family of dual specificity tyrosine kinases, like PNG is crucial for the oocyte-to-embryo transition. MBK-2 activation is linked to the meiotic cell cycle by being downstream of the APC/C (*Parry and Singson, 2011*). MBK-2 controls proteolysis of oocyte proteins through the SCF E3 ubiquitin ligase (*Robertson and Lin, 2013*), and it also affects RNA granule dynamics and thus likely impacts translation (*Wang et al., 2014*). Although PNG and MBK-2 are distinct kinases, these distantly related invertebrates utilize parallel approaches of coupling to cell cycle regulators to limit kinase activity to the oocyte-to-embryo transition. In both organisms, this links meiotic progression to gene expression changes after egg activation. This conservation suggests that this strategy may be employed for the onset of mammalian development.

## Materials and methods

We use the term 'biological replicate' to mean a repeat of all of the steps of an experiment, for example isolation of the flies, recovery of oocytes, preparation and hybridization of an immunoblot. A 'technical replicate' would be a repeat of an immunoblot from a previously used protein extract or repeated embryo collections and stainings from flies produced by a single cross.

### Fly stocks and transgenic lines

*Oregon R* was used as the wild-type control. *y png$^{1058}$ w/FM6, gnu$^{305}$, twine$^{HB5}$*, and *mos* have been described (*Freeman et al., 1986*; *Schüpbach and Wieschaus, 1989*; *Fenger et al., 2000*; *Ivanovska et al., 2004*). Flies were maintained at 22 or 25°C on standard Drosophila cornmeal molasses food.

To construct a *gnu-gfp* transgene driven by the genomic promoter, the genomic sequence including *gnu* was amplified by PCR with primers (TCGCTTTGTGCCCAGCCATC; AGTGTGACCGCGGA TCGACT) and cloned into pGEM T-easy (Promega, Madison, WI). The GFP sequence with a linker sequence (GGCGGAAGTGGAGCGGCCGCC) was inserted into the 3' of the *gnu* ORF using Gibson Assembly Master Mix (NEB, Ipswich, MA). For the phosphomutant, the *gnu* 9A mutant sequence was swapped with the sequence in the plasmid by Gibson Assembly. An approximately 3 kb genomic region with *gnu-gfp WT* or *9A* was cloned into pCasPeR4 (*Thummel and Pirrotta, 1992*), digested with *Bam*HI and *Eco*RI, by Gibson Assembly. pCasPeR4 *gnu-gfp WT* and pCasPeR4 *gnu-gfp 9A* were injected into *w$^{1118}$* embryos and transgenics were recovered by P element transposition (BestGene, Chino Hills, CA).

### Oocyte and embryo collection

Staged oocytes (egg chambers) were hand dissected from fattened females in Grace's Insect Medium, unsupplemented (Life Technologies, Carlsbad, CA). Fertilized embryos were collected for 2 hr, dechorionated in 50% bleach and washed with embryo wash buffer (0.9% NaCl, 0.03% Triton X-100). Unfertilized activated eggs were collected for 2 hr from females mated to sterile *twine$^{HB5}$* males (*Courtot et al., 1992*), dechorionated and washed as above.

For immunoblotting, thirty oocytes, unfertilized activated eggs or embryos in 15 μL medium or embryo wash buffer, were frozen in liquid $N_2$ in 1.5 mL tubes and stored at −80°C. After addition of 15 μL 2X Laemmli sample buffer (LSB), they were homogenized on ice and boiled for 5 min.

For immunofluorescence, activated eggs and embryos were fixed and stained with DAPI as in (*Shamanski and Orr-Weaver, 1991*). Mature oocytes were fixed and stained as in (*Page and Orr-Weaver, 1997*), except DAPI was used. The samples were scored on a Nikon ECLIPSE Ti microscope with Plan Fluor 10x or Plan Apo 20x objectives.

## Assay for phosphatase sensitivity of GNU

To examine phosphatase sensitivity of GNU, fifty stage 14 oocytes in 5 µL Grace's Insect Medium, unsupplemented (Life Technologies, Carlsbad, CA) were homogenized in 20 µL lysis buffer [50 mM Tris-HCl pH 8.0, 150 mM NaCl, 2.5 mM EGTA pH 8.0, 1% NP40, disodium $\beta$- glycerophosphate, 25 mM NaF, 1X Complete EDTA-free protease inhibitor cocktail (Roche, Indianapolis, IN)] in the presence or absence of 2.5 mM EDTA, with or without 100 nM okadaic acid. After spinning at 13.3 krpm at 4°C for 15 min, 10 µL of supernatant was mixed with 10 µL 2X Laemmli Sample Buffer (LSB) and boiled for 5 min. As a control, fifty stage 14 oocytes in 5 µL Grace's Insect Medium were homogenized in 45 µL 1X LSB and boiled for 5 min.

## In vitro egg activation

Stage 14 oocytes were activated as previously described (*Mahowald et al., 1983*; *Page and Orr-Weaver, 1997*; *Horner and Wolfner, 2008a*). Stage 14 oocytes were isolated in isolation buffer from virgin females enriched for mature oocytes (*Mahowald et al., 1983*) and activated in activation buffer. When activated oocytes were collected at more than 25 min after initiation of activation, the oocytes were washed with $H_2O$, transferred and incubated further in modified Zalokar's buffer [9 mM $MgCl_2$, 10 mM $MgSO_4$, 2.9 mM $NaH_2PO_4$, 0.22 mM NaOAc, 5 mM glucose, 34 mM glutamic acid, 33 mM glycine, 2 mM malic acid, 7 mM $CaCl_2$, pH 6.8 (1:1 NaOH:KOH)]. Activated oocytes were dechorionated and selected for successful activation by treatment with 50% bleach then washed with $H_2O$ and embryo wash buffer. Fixation and DAPI staining were as described (*Shamanski and Orr-Weaver, 1991*). The meiotic stages were defined by chromosome structure (*Page and Orr-Weaver, 1997*).

## Single in vitro activated oocyte assay

To examine protein dynamics in single oocytes in particular meiotic stages, we applied a published protocol that was used previously for embryos (*Edgar et al., 1994*). In vitro activated oocytes were dechorionated in 50% bleach, devitellinized and fixed in a bilayer of heptane and methanol for 5 min, rinsed three times with ice cold methanol and twice with EB [10 mM Tris-HCl pH 7.5, 80 mM disodium $\beta$-glycerophosphate, 20 mM EGTA pH 8.0, 15 mM $MgCl_2$, 1X Complete EDTA-free protease inhibitor cocktail (Roche, Indianapolis, IN), 1X PhosSTOP (Roche, Indianapolis, IN)] with 0.05% Tween-20, and the DNA stained with 4 µg/mL Hoechst 33342 in EB. After two rinses with EB, oocytes were incubated in 40%EB/60%Glycerol on ice for 1 hr and stored at −80°C. The oocytes were staged and selected using a fluorescence microscope (ECLIPSE Ti, Nikon Instruments, Melville, NY) with Plan Fluor 10x/0.30 (Nikon Instruments, Melville, NY) and Plan Apo 20x/0.75 (Nikon Instruments, Melville, NY) objectives. The meiotic stages were defined by chromosome structure (*Page and Orr-Weaver, 1997*). The images were acquired with a Plan Apo 20x/0.75 (Nikon Instruments, Melville, NY) objective using NIS Elements (Nikon Instruments, Melville, NY, RRID:SCR_014329) and processed with Fiji (*Schindelin et al., 2012*) and Photoshop (Adobe, San Jose, CA, RRID:SCR_014199). The late ana-II and telo-II images in *Figure 1D* were Z projection images processed by Fiji (*Schindelin et al., 2012*). After the image acquisition, each oocyte was recovered with 2 µL 40%EB/60%Glycerol into a tube, homogenized in 3 µL 4X LSB and boiled for 5 min.

## Immunoblots

Samples were separated by 7.5% or 10% SDS-PAGE and transferred to Immobilon polyvinylidene fluoride membranes (EMD Millipore, Billerica, MA). Proteins were detected by ECL2 (Thermo Fisher Scientific, Waltham, MA) and visualized with ChemiDoc XRS+ (BioRad, Hercules, CA). To reprobe with another antibody, the membranes were stripped with Restor Plus Western Blot Stripping Buffer (Thermo Scientific, Waltham, MA).

Primary antibodies used were rabbit anti-PNG (this study), rabbit anti-PLU (*Elfring et al., 1997*), guinea pig anti-GNU (*Lee et al., 2003*), guinea pig anti-SMG (from Craig Smibert, University of Toronto), mouse anti-CycA (Developmental Studies Hybridoma Bank, Iowa City, IA, RRID:AB_528188) and anti-CycB (Developmental Studies Hybridoma Bank, Iowa City, IA, RRID:AB_528189), guinea pig anti-GFP (a gift from Mary-Lou Pardue, MIT), rat anti-α tubulin YOL1/34 (AbD Serotec, Raleigh, NC, RRID:AB_325005), rabbit anti-GST labeled with HRP (MBL, Woburn, MA, RRID:AB_591785) and mouse anti-FLAG M2 (Sigma-Aldrich, St. Louis, MO, RRID:AB_439685). Secondary antibodies were

HRP-conjugated anti-rabbit IgG (Jackson ImmunoResearch, West Grove, PA, RRID:AB_10015282), anti-guinea pig IgG (Jackson ImmunoResearch, West Grove, PA, RRID:AB_2340447), anti-mouse IgG (Jackson ImmunoResearch, West Grove, PA, RRID:AB_2338510) and anti-rat IgG (Jackson ImmunoResearch, West Grove, PA, RRID:AB_2338133). We used Signal Enhancer Hikari (Nacalai USA, San Diego, CA) to increase sensitivity and specificity of the antibodies for PNG, PLU, SMG, GFP, CycA, and CycB.

Immunoblots to examine levels of the PNG, GNU, SMG, Cyclin B, or Cyclin A protein levels during development were repeated in at least two biological replicates.

## PNG antibody

His-tagged full length PNG protein was expressed in Rosetta 2(DE3) (EMD Millipore, Billerica, MA) using pET28b (EMD Millipore, Billerica, MA) and purified with Ni-NTA Agarose (Qiagen, Valencia, CA) under denaturing condition. Purified His-PNG was injected to rabbits to raise antibodies (Covance, Princeton, NJ). The specificity of the antibody was confirmed by western blots of *png* mutant extracts.

## Immunoprecipitation for GNU phosphomapping

Stage 14 oocytes were isolated in isolation buffer (*Page and Orr-Weaver, 1997*) from female flies expressing a GNU-GFP-WT and frozen in liquid nitrogen. The oocytes were briefly homogenized in IP-lysis buffer [25 mM HEPES pH 7.5, 150 mM NaCl, 20 mM EGTA pH8.0, 15 mM MgCl$_2$, 1 mM DTT, 0.5% NP-40, 10% glycerol, 1X Complete EDTA-free protease inhibitor cocktail (Roche, Indianapolis, IN), 2X PhosSTOP (Roche, Indianapolis, IN), 250 nM okadaic acid] and sonicated in a Bioruptor (Diagenode) at 4°C for 5 cycles at 30 s on and 30 s off. After spinning at 13.3 krpm at 4°C for 5 min, supernatants were transferred to new tubes and the protein concentration adjusted to 1 µg/µL. GNU-GFP was immunoprecipitated from 1.3 mg of the protein extracts with 35 µL GFP-TRAP_MA beads (Chromotek, Planegg-Martinsried, Germany) for 10 min. After three washes with IP-lysis buffer, proteins were eluted by boiling in 4X LSB for 5 min.

## Sample preparation for mass spectrometry

Duplicate coomassie stained SDS–PAGE gels were run to obtain separate gel bands for digestion with the endoproteases trypsin and chymotrypsin. Bands representing GNU as well as representative blank controls were excised and cut into ~2 mm squares. These were washed overnight in 50% methanol/ water. These were washed once more with 47.5/47.5/5% methanol/water/acetic acid for 2 hr, dehydrated with acetonitrile and dried in a speed-vac. Reduction and alkylation of disulfide bonds was then carried out by the addition of 30 µL 10 mM dithiothreitol (DTT) in 100 mM ammonium bicarbonate for 30 min. The resulting free cysteine residues were subjected to an alkylation reaction by removal of the DTT solution and the addition of 100 mM iodoacetamide in 100 mM ammonium bicarbonate for 30 min to form carbamidomethyl cysteine. These were then sequentially washed with aliquots of acetonitrile, 100 mM ammonium bicarbonate and acetonitrile and dried in a speed-vac. The bands were enzymatically digested by the addition of 300 ng of trypsin or chymotrypsin in 50 mM ammonium bicarbonate to the dried gel pieces for 10 min on ice. Depending on the volume of acrylamide, excess ammonium bicarbonate was removed or enough was added to rehydrate the gel pieces. These were allowed to digest overnight at 37°C with gentle shaking. The resulting peptides were extracted by the addition of around 50 µL of 50 mM ammonium bicarbonate with gentle shaking for 10 min. The supernatant from this was collected in a 0.5 ml conical autosampler vial. Two subsequent additions of 47.5/47.5/5% acetonitrile/water/formic acid with gentle shaking for 10 min were performed with the supernatant added to the 0.5 ml autosampler vial. Organic solvent was removed and the volumes were reduced to 15 µL using a speed-vac for subsequent analyses.

## Chromatographic separation of peptides

The digestion extracts were analyzed by reversed phase high performance liquid chromatography (HPLC) using Waters NanoAcquity pumps and autosampler (Waters, Milford, MA) and a Thermo-Fisher Orbitrap Elite mass spectrometer (ThermoFisher Scientific, Waltham, MA) using a nano flow configuration. A 20 mm x 180 micron column packed with five micron Symmetry C18 material

(Waters, Milford, MA) using a flow rate of 15 µL/min for 3 min was used to trap and wash peptides. These were then eluted onto the analytical column which was a self-packed with 3.6 micron Aeris C18 material (Phenomenex, Torence, CA) in a fritted 20 cm x 75 micron fused silica tubing pulled to a five micron tip. The gradient was isocratic 1% A Buffer for 1 min 250 nL/min with increasing B buffer concentrations to 15% B at 20.5 min, 27% B at 31 min and 40% B at 36 min. The column was washed with high percent B and re-equilibrated between analytical runs for a total cycle time of approximately 53 min. Buffer A consisted of 1% formic acid in water and buffer B consisted of 1% formic acid in acetonitrile.

## Mass spectrometry and data analysis

The mass spectrometer was operated in a data dependent acquisition mode, where the 10 most abundant peptides detected in the Orbitrap using full scan mode with a resolution of 240,000 were subjected to daughter ion fragmentation in the linear ion trap using multi stage activation where neutral losses of 97.97, 48.99 or 32.66 m/z below the precursor were fragmented further (pseudo MS3). A running list of parent ions was tabulated to an exclusion list to increase the number of peptides analyzed throughout the chromatographic run.

Peptides were identified from the MS data using PEAKS Studio 8.0 (Bioinformatics Solutions), Mascot (Matrix Science, Boston, MA, RRID:SCR_014322) and Sequest (ThermoFisher Scientific, Waltham, MA) algorithms. Species-specific (*Drosophilia melanogaster*) Refseq FASTA files were downloaded from NCBI and concatenated to a database of common contaminants (keratin, trypsin, etc). The resulting search results from PEAKS, Mascot and Sequest were then loaded into Scaffold (Proteome Software, Portland, OR, RRID:SCR_014345). Phosphopeptides with an Ascore above 10 were considered positive identifications (*Beausoleil et al., 2006*).

## Purified recombinant proteins

The *gnu* ORF was cloned into pGEX6P-1 (GE Healthcare, Waukesha, WI). A CDK1 site mutant (9A) in which Ser13, Thr16, Thr19, Thr32, Ser92, Ser156, Thr158, Ser170 and Ser178 are substituted by Ala was synthesized (GENEWIZ, South Plainfield, NJ). *gnu 9A* also was cloned into pGEX6P-1. GST-GNU proteins were expressed in BL21 and purified by glutathione sepharose 4B (GE Healthcare, Waukesha, WI).

PNG kinase complex was expressed and purified from Sf9 insect cells. *png-FLAG* and *plu-His* sequence were cloned in multiple cloning sites downstream of the p10 promoter and polyhedrin promoter, respectively, in pFastBac Dual (Life Technologies, Carlsbad, CA). *gnu WT* was cloned into pFastBac1 (*Lee et al., 2003*). Recombinant baculoviruses were generated, amplified and used to infect Sf9 cells according to the instructions for the Bac-to-Bac Baculovirus expression system (Life Technologies, Carlsbad, CA). Seventy-two hours after infection, cells were harvested, washed with ice cold PBS, lysed in Sf9 lysis buffer [25 mM HEPES-NaOH pH 7.8, 1 mM EGTA pH 8.0, 150 mM NaCl, 80 mM disodium β- glycerophosphate, 1 mM DTT, 10% glycerol, 1X Complete EDTA free protease inhibitor cocktail (Roche, Indianapolis, IN)] with 1% NP-40, and spun to recover the soluble fraction. The PNG complex was bound to FLAG M2 Affinity Gel (Sigma-Aldrich, St. Louis, MO), and after washing with Sf9 lysis buffer with 0.05% NP-40, eluted by Sf9 lysis buffer with 0.05% NP-40 and 500 µg/mL FLAG peptide (Sigma-Aldrich, St. Louis, MO).

## PNG complex formation assay in Sf9 cells

*gnu WT* or *9A* were cloned into pFastBac1-His-HA (*Lee et al., 2003*). Recombinant baculoviruses were generated as above. Sf9 cells were cotransfected with *gnu-His-HA and png-FLAG* and *plu-His* viruses. Seventy-two hour after infection, cells were treated with or without 100 nM okadaic acid for 12 hr. Then, cells were washed with ice cold PBS, harvested, lysed in Sf9 lysis buffer with 1% NP-40, and spun to recover the soluble fraction. The PNG kinase complex was immunoprecipitated with FLAG M2 Affinity Gel (Sigma-Aldrich, St. Louis, MO). After washing the gel with Sf9 lysis buffer supplemented with 1% NP-40, the proteins were eluted by adding 2X LSB and boiled for 5 min.

This experiment was repeated in three biological replicates.

## Histone H1 kinase assay

Twenty staged oocytes or in vitro activated oocytes were recovered in 6 μL Grace's Insect Medium, unsupplemented (Life Technologies, Carlsbad, CA) or embryo wash buffer and frozen in Liquid $N_2$. After addition of 34 μL H1K lysis buffer [20 mM Tris-HCl pH 7.5, 80 mM disodium $\beta$-glycerophosphate, 20 mM EGTA pH 8.0, 15 mM $MgCl_2$, 1 mM DTT, 1X Complete EDTA-free protease inhibitor cocktail (Roche, Indianapolis, IN), 1X PhosSTOP (Roche, Indianapolis, IN)], the egg chambers and the in vitro activated oocytes were homogenized and spun at 13.3 krpm at 4°C for 15 min. One microliter of the supernatants was incubated with 9 μL of an H1K reaction mixture (80 mM disodium $\beta$-glycerophosphate, 20 mM EGTA, 15 mM $MgCl_2$, 1 mM DTT, 0.6 mg/mL Histone H1, 10 μM ATP, 14.8 MBq/ml [$\gamma$-$^{32}$P]ATP, pH 7.3) at 25°C for 15 min. The reaction was stopped by addition of 5 μL 4X LSB and boiled for 5 min. Following SDS-PAGE, after Coomassie Brilliant Blue (CBB) staining phosphorylated Histone H1 was detected by autoradiography.

## GNU phosphorylation by CycB/CDK1

Active CycB/CDK1 purified from Starfish oocytes [a gift from Takeo Kishimoto and Ei-ich Okumura; (Okumura et al., 1996)] was used for GNU phosphorylation assay. CycB/CDK1 (7.5 pmol P/min) was incubated 0.2 μg recombinant GST-GNU WT or 9A in 10 μL CDK1 reaction buffer [10 mM Tris-HCl pH 7.5, 2 mM $MgCl_2$) with 100 μM ATP and 1.85 MBq/mL [$\gamma$-$^{32}$P]ATP] at 25°C for 30 min, then 5 μL 3X LSB with 25 mM EDTA was added and the sample boiled for 5 min. Samples were separated on 10% SDS-PAGE, and after CBB staining phosphorylated GNU was detected by autoradiography.

To prepare CDK1-phosphorylated GNU, 30 μg GST-GNU and CycB/CDK1 (750 pmol P/min) were incubated in 250 μL CDK1 reaction buffer with 1 mM ATP at 25°C for 1 hr. Phosphorylated GNU was supplemented with 150 mM NaCl, 1 mM DTT and 0.05% NP-40, purified with glutathione sepharose 4B (GE Healthcare, Waukesha, WI) and dialyzed to TBS with 1 mM DTT and 0.05% NP-40.

## PNG kinase activation assay

Recombinant PNG kinase complex containing 6 ng PNG-FLAG was incubated with varying amounts (0–80 ng) of GST-GNU, phosphorylated or unphosphorylated by CycB/CDK1, in 10 μL PNG reaction buffer [20 mM Tris-HCl pH 7.5, 3 mM $MnCl_2$, 10 mM $MgCl_2$, 80 mM disodium $\beta$-glycerophosphate, 100 μM ATP, 1 mM DTT, 1X Complete EDTA-free protease inhibitor cocktail (Roche, Indianapolis, IN)] with 10 μM Roscovitine, 7.4 MBq/mL [$\gamma$-$^{32}$P]ATP and 0.6 mg/mL myelin basic protein (MBP, a PNG kinase in vitro substrate) at 30°C for 15 min. 5 μL 3X LSB with 25 mM EDTA was added, and the sample boiled for 5 min. GST-GNU, phosphorylated or unphosphorylated by CycB/CDK1, was also examined without PNG kinase complex as a negative control. Samples were separated on 15% SDS-PAGE and after CBB staining phosphorylated MBP was detected by autoradiography.

## PNG kinase activation assay with GNU mutants

To examine whether GNU 9A activates PNG kinase in vitro, recombinant PNG kinase complex containing 6 ng PNG-FLAG was incubated with 20 ng GST-GNU WT, 9A or P17A (a negative control) in 10 μL PNG reaction buffer [20 mM Tris-HCl pH7.5, 3 mM $MnCl_2$, 10 mM $MgCl_2$, 80 mM disodium $\beta$-glycerophosphate, 100 μM ATP, 1 mM DTT, 1X complete EDTA-free (Roche, Indianapolis, IN)] with 7.4 MBq/mL [$\gamma$-$^{32}$P]ATP and 0.6 mg/mL myelin basic protein (MBP, a PNG kinase in vitro substrate) at 30°C for 15 min, then 5 μL 3X LSB with 25 mM EDTA was added and the sample boiled for 5 min. Samples were separated on 15% SDS-PAGE, and after CBB staining phosphorylated MBP was detected by autoradiography. Radioactivity incorporated into MBP was quantified by liquid scintillation counting.

## Immunoprecipitation

Stage 14 oocytes were isolated in Grace's Unsupplemented Insect Medium (Life Technologies, Carlsbad, CA) from female flies expressing a GNU-GFP-WT or -9A transgene, washed with embryo wash buffer and frozen in liquid nitrogen. The oocytes were homogenized in IP-lysis buffer [25 mM HEPES pH 7.5, 150 mM NaCl, 20 mM EGTA pH8.0, 15 mM $MgCl_2$, 1 mM DTT, 0.5% NP-40, 10% glycerol, 1X Complete EDTA-free protease inhibitor cocktail (Roche, Indianapolis, IN), 1X PhosSTOP (Roche, Indianapolis, IN), 125 nM okadaic acid]. After spinning at 13.3 krpm at 4°C for 15 min, supernatants were transferred to new tubes and the protein concentration adjusted to 0.25 μg/μL. PNG was

immunoprecipitated from 25 µg of the protein extracts with Protein A sepharose CL4B (GE Healthcare, Waukesha, WI) crosslinked with PNG serum by BS[3] (Thermo Fisher Scientific, Waltham, MA). After three washes with IP-lysis buffer, proteins were eluted with 2X LSB and boiled for 5 min.

This experiment was repeated in two biological replicates.

## Phosphatase treatment of immunoprecipitated PNG complexes

To examine GNU protein levels in the soluble fraction and immunoprecipitated PNG complexes on immunoblot precisely, they were dephosphorylated with λ-phosphatase (NEB, Ipswich, MA) to be condensed mobility shifted GNU bands via phosphorylation to a single band. Briefly, the soluble fractions were supplemented to a final concentration of 1 mM $MnCl_2$ and incubated with λ-phosphatase at 30°C for 30 min. The reaction was stopped by addition of 2xLSB and boiling for 5 min. For the immunoprecipitated PNG complexes, they were washed with λ-phosphatase buffer [50 mM HEPES pH 7.5 (NaOH), 10 mM NaCl, 1 mM $MnCl_2$, 2 mM DTT, 0.05% NP-40] and incubated with λ-phosphatase in λ-phosphatase buffer at 30°C for 30 min. The reaction was stopped by LSB addition as above.

## Acknowledgements

We are extremely grateful to Eric Spooner of the Whitehead Institute for help with the mass spec mapping of GNU phosphorylation sites. Jessica Von Stetina, Emir Aviles Pagan, and Helena Kashevsky provided assistance with transgenic stocks and embryo collections. We thank Takeo Kishimoto and Ei-ich Okumura for their generous gift of purified CycB/CDK1 kinase, Laura Lee for the GNU bacmid, and Iva Kronja for communicating unpublished results. We are grateful to Mary-Lou Pardue and Craig Smibert for antibodies. We thank Adam Martin for his comments on the manuscript, as well as Angelika Amon, Tony Mahowald, Emir Aviles Pagan, and Iva Kronja for helpful comments on early versions. This research was supported by a JSPS Postdoctoral Fellowship for Research Abroad and an Uehara Memorial Foundation Research Fellowship to MH and by NIH grants GM39341 and GM118098 to TO-W. TO-W is an American Cancer Society Research Professor.

## Additional information

### Funding

| Funder | Grant reference number | Author |
| --- | --- | --- |
| Japan Society for the Promotion of Science | | Masatoshi Hara |
| Uehara Memorial Foundation | | Masatoshi Hara |
| National Institutes of Health | GM39341 | Terry L Orr-Weaver |
| American Cancer Society | | Terry L Orr-Weaver |
| National Institutes of Health | GM118098 | Terry L Orr-Weaver |

The funders had no role in study design, data collection and interpretation, or the decision to submit the work for publication.

### Author contributions

MH, Conceptualization, Data curation, Formal analysis, Validation, Investigation, Visualization, Methodology, Writing—original draft, Writing—review and editing; BP, Conceptualization, Data curation, Formal analysis, Validation, Investigation, Visualization, Methodology, Writing—review and editing; TLO-W, Conceptualization, Data curation, Formal analysis, Supervision, Funding acquisition, Validation, Investigation, Visualization, Methodology, Writing—original draft, Project administration, Writing—review and editing

### Author ORCIDs

Masatoshi Hara, http://orcid.org/0000-0001-8433-1111
Terry L Orr-Weaver, http://orcid.org/0000-0002-7934-111X

### Ethics

Animal experimentation: The rabbits used and the protocol to produce the antibodies for this study followed the recommendations in the Guide for the Care and Use of Laboratory Animals of the National Institutes of Health. The protocols were approved by the institutional animal care and use committee (IACUC) of the Massachusetts Institute of Technology, protocol numbers 0310-024-13 and 0313-020-16.

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
