## [Decision Letter]

Thank you for submitting your article "Control of PNG kinase by phosphorylation links mRNA translation to meiosis completion at the oocyte-to-embryo transition" for consideration by *eLife*. Your article has been favorably evaluated by Kevin Struhl (Senior Editor) and three reviewers, one of whom, Jon Pines (Reviewer #1), is a member of our Board of Reviewing Editors.

The reviewers have discussed the reviews with one another and the Reviewing Editor has drafted this decision to help you prepare a revised submission. We realise that the work required will take longer than the usual 2 months that we prefer for revising a manuscript but if you think it will take more than 6 months please let us know. Please also inform us if you prefer not to carry out the work and submit to another journal.

In this study the authors have investigated the control of maternal mRNA translation at the oocyte to egg transition in *Drosophila*. They show that all the components of the PNG-PLU kinase are present but inactive while Cyclin B kinase levels are high. They show that this is because cyclin B-Cdk1 phosphorylates the GNU protein and that this prevents PNG binding to PLU. Once Cyclin B-Cdk1 activity declines, GNU is dephosphorylated and PNG kinase activated, which leads to a change in mRNA translation and a reduction in GNU levels. A non-phosphorylatable form of GNU is only partially active in vitro and in vivo, but sufficient to show that preventing its regulation by cyclin B-Cdk1 leads to premature PNG activation and disruption of meiosis. Overall the results described in this paper are clear and convincing and make an important contribution to our knowledge of the oocyte to egg transition that is appropriate for publication in *eLife*. The experiments are well designed and appropriate controls included.

Essential revisions:

What is missing is a proper analysis of the phosphorylation of GNU. The authors rely on mobility shifts and mutate all the potential Cdk1 consensus sites. This 9x alanine mutant could have properties unrelated to resistance to Cdk1 phosphorylation and therefore be misleading. More contemporary methods such as mass spectrometry will identify the exact sites and allow the authors to test their effect in vivo. Without these data the study is incomplete and will not be acceptable for *eLife*.

In addition to these data the authors should also address the following issues:

1) Title: the authors have not investigated the control of translation and should remove this from the title.

2) Two independent transformant lines are shown in Figure 4. The lines should be genetically identical, but they give different results. Line 2-7 gives the effect on *cycB* levels described by the authors. However, line 2-8's results look the like the control's. Please explain this discrepancy, and perhaps modify the paper's conclusions in response.

3) The authors say that *cycB* levels were "significantly higher" in line 2-7, but no p-value is provided, and it is not easy to be certain of statistical significance given the overlapping error bars. Please provide statistical information.

4) It would help to say a bit more about the nature of the chromosomal aberrations seen in vivo, so that readers can have a more complete idea of what is wrong, or what is not seen. Also, perhaps discuss the reason for the many non-aberrant nuclei in the mutant.

---

## [Author Response]

*Essential revisions:*

*What is missing is a proper analysis of the phosphorylation of GNU. The authors rely on mobility shifts and mutate all the potential Cdk1 consensus sites. This 9x alanine mutant could have properties unrelated to resistance to Cdk1 phosphorylation and therefore be misleading. More contemporary methods such as mass spectrometry will identify the exact sites and allow the authors to test their effect* in vivo*. Without these data the study is incomplete and will not be acceptable for eLife.*

We used mass spectrometry to map the phosphorylated amino acids in GNU immunoprecipitated from mature oocytes. These data are now included in the revised Figure 2 and Figure 2—figure supplement 1. We found that GNU is phosphorylated in vivo on the nine predicted Cdk1 sites. Four other phosphorylated residues were detected less frequently. Given that in vivo GNU is phosphorylated on all nine Cdk1 sites, analysis of the properties of the GNU-9A phospho-resistant form of the protein in vitro and expression of this form of the protein in mature oocytes was the appropriate experimental approach.

*In addition to these data the authors should also address the following issues:*

1) Title: the authors have not investigated the control of translation and should remove this from the title.

We changed the title as suggested.

*2) Two independent transformant lines are shown in Figure 4. The lines should be genetically identical, but they give different results. Line 2-7 gives the effect on cycB levels described by the authors. However, line 2-8's results look the like the control's. Please explain this discrepancy, and perhaps modify the paper's conclusions in response.*

These transgenic lines contain the *gnu* gene and genomic regulatory regions sufficient for rescue, and they are inserted at ectopic sites in the genome. Consequently, expression levels can differ between transgenic lines. As shown in Figure 4, the two wild-type lines and the 2-7 line have comparable levels of GNU protein, but levels are slightly lower in the 2-8 line. This is now explained in the text.

*3) The authors say that cycB levels were "significantly higher" in line 2-7, but no p-value is provided, and it is not easy to be certain of statistical significance given the overlapping error bars. Please provide statistical information.*

The experiment to quantify Cyclin B protein levels in the mature oocytes was repeated in five independent biological replicates. Both the 2-7 and 2-8 transgene lines have significantly higher levels of Cyclin B protein by a T test (P=0.03 for line 2-7 and P=0.01 for line 2-8). The quantification of the immunoblots is presented in source files.

*4) It would help to say a bit more about the nature of the chromosomal aberrations seen* in vivo*, so that readers can have a more complete idea of what is wrong, or what is not seen. Also, perhaps discuss the reason for the many non-aberrant nuclei in the mutant.*

Additional details on the aberrant chromosome configurations are provided, as is a further explanation of the unequal segregation of meiotic chromosomes. The limitation of cytology given that in *Drosophila* there are only four chromosomes, two of which are the same size, is noted. It is explicitly stated that as a consequence the error rate is an underestimate.